# Efficacy of signal peptide predictors in identifying signal peptides in the experimental secretome of *Picrophilous torridus*, a thermoacidophilic archaeon

**Neelja Singhal[1], Anjali Garg[1], Nirpendra Singh[2], Pallavi Gulati[3], Manish Kumar[1], Manisha Goel** [1] *

**1** Department of Biophysics, University of Delhi South Campus, New Delhi, India, **2** Regional Center for Biotechnology, NCR-Biotech Science Cluster, Faridabad, India, **3** Department of Microbiology, University of Delhi South Campus, New Delhi, India

* manishagoel@south.du.ac.in

**Data Availability Statement:** All relevant data are within the manuscript and its Supporting Information files.

## Abstract

Secretory proteins are important for microbial adaptation and survival in a particular environment. Till date, experimental secretomes have been reported for a few archaea. In this study, we have identified the experimental secretome of *Picrophilous torridus* and evaluated the efficacy of various signal peptide predictors (SPPs) in identifying signal peptides (SPs) in its experimental secretome. Liquid chromatography mass spectrometric (LC MS) analysis was performed for three independent *P. torridus* secretome samples and only those proteins which were common in the three experiments were selected for further analysis. Thus, 30 proteins were finally included in this study. Of these, 10 proteins were identified as hypothetical/uncharacterized proteins. Gene Ontology, KEGG and STRING analyses revealed that majority of the secreted proteins and/or their interacting partners were involved in different metabolic pathways. Also, a few proteins like malate dehydrogenase (Q6L0C3) were multi-functional involved in different metabolic pathways like carbon metabolism, microbial metabolism in diverse environments, biosynthesis of antibiotics, etc. Multi-functionality of the secreted proteins reflects an important aspect of thermoacidophilic adaptation of *P. torridus* which has the smallest genome (1.5 Mbp) among nonparasitic aerobic microbes. SPPs like, PRED-SIGNAL, SignalP 5.0, PRED-TAT and LipoP 1.0 identified SPs in only a few secreted proteins. This suggests that either these SPPs were insufficient, or N-terminal SPs were absent in majority of the secreted proteins, or there might be alternative mechanisms of protein translocation in *P. torridus*.

## Introduction

*Picrophilus torridus* is an extremely acidophilic and moderately thermophilic (optimal growth temperature ~ 55–60˚C) euryarchaeon, which was first isolated from dry solfataric fields of northern Japan [1]. The whole genome sequence analysis of *P. torridus*

**Funding:** The author(s) received no specific funding for this work.

**Competing interests:** The authors have declared that no competing interests exist.

revealed that it had the highest coding density among thermoacidophiles and the smallest genome (1.55 Mbp) among nonparasitic aerobic microbes culturable on organic substrates [2]. Though the intracellular pH of thermoacidophiles is reportedly *ca*. neutral, but in case of *Picrophilus* spp. an unusual intracellular pH of around 4.6 has been reported [3].

Microbial secretome/secreted proteins play an important role in adaptation and survival in a particular niche, including thermoacidophilic environment. The secretome performs a variety of functions like degradation of complex polymeric substances (carbohydrates and proteins), passage of nutrients inside the cell, protection against toxic compounds, signal transduction *etc* [4,5]. In prokaryotes, eukaryotes and archaea, a variety of transport systems can be utilized for protein secretion. The ABC transporters are used for excretion of peptides and toxins [6]. The universally conserved general secretory pathway (Sec-pathway) is used for translocation of unfolded secretory proteins across the cytoplasmic membrane [7]. The proteins intended for secretion harbour a signal peptide (SP) at their N-terminal. The SP is made up of three regions: the N- terminal (n-region) containing positively charged amino acid residues, the hydrophobic (h-) region containing hydrophobic amino acid residues and a c-region containing small, uncharged amino acid residues and a characteristic cleavage site [8]. The SPs are cleaved from the proteins during or after their translocation across the cell membrane by specialized enzymes called signal peptidases. The signal peptidases are of two types, signal peptidase I (SPase I) or signal peptidase II (SPase II). SPase I substrates are usually released as soluble proteins, whereas SPase II substrates (lipoproteins) are attached to the cell membrane with the help of a lipid anchor. Although, genomes of many archaea encode for proteins whose N-terminal contain lipobox, SPase II homologs are detected rarely in archaea [9]. Apart from the Sec-pathway, the twin-arginine translocation (TAT) pathway is another protein translocation pathway which allows secretion of folded proteins [10]. The TAT substrates were reportedly present in haloarchaea like *Haloferax volcanii* and *Natrinema* sp. J7-2 [11,12].

A very few studies have investigated the composition of archaeal secretomes. To the best of our knowledge, till date, experimental secretomes have been identified for an antartic archaeon *Methanococcoides burtonii* [13], hyperthermoacidophilic archaeon *Sulfolobus* spp. [14], hyperthermophilic archaeon *Pyrococcus furiosus* [5] and haloarchaea like *Haloferax volcanii* and *Natrinema* sp. J7-2 [11,12]. However, secreted proteins of thermoacidophilic archaeon *P. torridus* have not been identified experimentally, till date. An earlier study, reported the composition of whole cell proteins of *P. torridus* using a bottom down proteomics approach, where proteins separated by two-dimensional (2D) gel electrophoresis were identified by mass spectrometry [15]. In the present study we have discerned the experimental secretome of *P. torridus* using liquid chromatography mass spectrometry (LC MS) and evaluated the efficacy of four signal peptide predictors (SPPs)—PRED-SIGNAL [16], SignalP 5.0 [17], PRED-TAT [18] and LipoP 1.0 [19] in identifying SPs in the experimental secretome. Though, many SPPs are available for predicting SPs like, SignalP 4.0 [20], Phobius [21], DeepSig [22] *etc*., in this study only four SPPs were used. PRED-SIGNAL was used because it was specifically designed for prediction of archaeal SPs and was trained on archaeal proteins having experimentally verified SPs [16]. The reason underlying the use of SignalP 5.0 was that, besides being one of the most cited and widely used SPPs, SignalP 5.0 can predict SPs and their cleavage sites in archaeal proteins, also [17]. Since, earlier studies have reported that TAT substrates and lipoproteins were abundant in the secretome of archaea [5,13] hence; SPPs PRED-TAT and LipoP, respectively [18,19] were used to discern their presence in *P. torridus* secretome.

## Materials and methods

### Bacterial culture and growth conditions

*P. torridus* (DSM 9790) was purchased from Leibniz Institute, DSMZ-German Collection of Microorganisms and Cell Cultures GmbH, Germany. The archaeal cells were grown at 55°C in 1 L of the culture medium in a shaking incubator set at 100 rpm. The components of the culture medium were: 0.05% magnesium sulfate, 0.025% calcium chloride, 0.02% ammonium sulfate, 0.3% potassium dihydrogen phosphate, 0.2% yeast extract and 1% glucose; pH 1.0 [23].

### Preparation of culture filtrate proteins

The *P. torridus* cultures were sampled at late exponential growth phase (~1–2 x $10^8$ cells/ml). The cultures were transferred to 500 ml centrifuge bottles and centrifuged at 8000 rpm at 4°C for 30 min. The supernatant was sterile filtered using filters of 0.2 μm pore size. The proteins in the cell free supernatant were concentrated using a Vivacell 250 ultra-filtration unit (Sartorius AG, Germany; filter cut off 3 kDa) following the manufacturer's instructions, to a final volume of 0.5 ml.

### Identification of culture filtrate proteins by LC MS

The concentrated culture filtrate proteins were precipitated with 10% trichloroacetic acid (TCA) at 4°C, overnight. The resulting pellet was processed for protein identification by LC MS using the methods described earlier [24]. Briefly, the protein pellet was washed with sodium acetate solution (2% in ethanol), kept for air-drying and finally resuspended in 200 μl of 8 M urea buffer (UB). Then 100 μl of 0.05 M iodoacetamide (IAA) was added, kept for 20 min incubation and centrifuged, followed by two washes with 100 μl of 0.05 M ammonium bicarbonate (ABC) and, centrifugation. This was followed by addition of 40 μl of ABC with trypsin (Promega V511A) (enzyme: protein ratio 1:100) and incubation at 37°C in a water bath for 16–18 h. The digested peptides were eluted by centrifugation at $14,000 \times g$ for 10 min; acidified with 0.1% formic acid and finally concentrated to 10 μl using a speed vac. LC MS analysis of the secretome was performed using AB SCIEX Triple TOF 5600. The peptides were identified by the ProteinPilot software version 4.0 (AB SCIEX) using Paragon algorithm as the search engine. The proteins with a cut-off set at 1% false-discovery rate and a minimum of 2-peptide-per-protein were selected for further study. The LC MS analysis was performed for three independent *P. torridus* secretome samples and only those proteins which were common in the three experiments were selected for further analysis (S1–S3 Files).

### Gene ontology and protein-protein interaction (PPI) studies

The functional annotation of the secretome was performed using the slim version of Gene ontology (GO) terms retrieved from the Gene Ontology Consortium [25]. The information about the interactome of secretory proteins of *P. torridus* was retrieved from STRING (database version 10.5)—a public repository of protein-protein interaction networks [26]. The analysis parameters included data from all the interaction sources like text mining, experiments, databases, co-expression, neighbourhood at default values. Interacting partners of the proteins were discerned using an *in-house* perl script and a confidence value ≥0.4. An interaction network of the secreted proteins was constructed using Cytoscape version 3.6.1 [27]. Simultaneously, the secretory proteins were also mapped on their corresponding metabolic networks in the Kyoto Encyclopaedia of Genes and Genomes (KEGG). KEGG is an extensively used reference knowledge base that cross-integrates genomic, chemical, and systemic functional information of an organism [28].

## Computational analysis of hypothetical/uncharacterized proteins

Computational analysis of the probable function of hypothetical/uncharacterized proteins was done using BLASTp (http://blast.ncbi.nlm.nih). The top five BLAST hits were selected for annotating the function of each hypothetical protein. BLAST search was performed at NCBI using the default threshold E-value—10, including the threshold value of 0.005. The domains present in the hypothetical proteins were discerned using Conserved Domain Database (CDD) (https://www.ncbi.nlm.nih.gov/cdd), Pfam 32 (https://pfam.xfam.org/) and InterPro 74 (https://www.ebi.ac.uk/interpro/). The top five BLAST hits were selected for functional annotation and probing the conserved domains of each hypothetical/uncharacterized secretory protein.

## Identification of N-terminal signal sequences

The N-terminal signal sequences in the culture filtrate proteins were identified using the SPPs like PRED-SIGNAL, SignalP 5.0, PRED-TAT and LipoP 1.0. PRED-SIGNAL was a SPP which was trained on archaeal secretory proteins and was especially designed for identification of SPs in archaeal proteins [17]. SignalP 5.0 can predict the SPs and their cleavage sites in proteins of gram-positive and–negative bacteria, archaea and eukaryotes [18]. PRED-TAT is a SPP program which can predict twin-arginine and secretory SPs in proteins of both gram-positive and–negative bacteria [18]. The LipoP 1.0 is a SP prediction program which can discriminate between lipoprotein SPs, other SPs and N-terminal membrane helices in Gram-negative bacteria [19].

# Results

## LC MS based protein identification and discerning the domains in hypothetical proteins for functional annotation

The number of proteins identified by LC MS in the three independent experiments was 68, 75 and 97 (S1–S3 Files). Only 30 proteins which were found to be present in the three independent secretome samples of *P. torridus* were selected for further analysis. The details of the 30 proteins selected for further analysis are shown in **Table 1** and their prominent domains are depicted using a Circos plot (S1 Fig). Of the 30 proteins, the 3D structure of only malate dehydrogenease (Q6L0C3) was present in the Protein Data Bank (PDB). PDB BLAST of the other proteins revealed that only 18 proteins showed identity with the known 3D protein structures available in the PDB (S1 Table). Ten proteins were hypothetical/uncharacterized *viz*. Q6L2C5, Q6L2C8, Q6KZG4, Q6KZB9, Q6L1G3, Q6L2S5, Q6L2L9, Q6L268, Q6L1Y4 and Q6KZK5. The protein domains discerned in the hypothetical proteins using CDD, Pfam and InterPro are summarized briefly in **Table 2**. The top five BLAST hits of Q6L2C5 revealed that it was a conserved protein in *Picrophilus* spp. and other archaea like Thermoplasmatales archaeon I-plasma and *Aciduliprofundum* sp. MAR08-339. Pfam did not find any domain but InterPro predicted a domain of unknown function DUF929 (IPR009272), while CDD search predicted a domain of Reo_sigmaC super family. The top four hits of Q6L2C8 indicated that it was a Von Willebrand factor type A (VWA)-domain containing protein also reported in archaea like *Sulfurisphaera tokodaii* and *Acidianus* spp. One BLAST hit indicated that it was a hypothetical protein of archaea *Candidatus aramenus sulfurataquae*. InterPro and Pfam revealed presence of VWA domain and archaellum regulatory network B, C-terminal. CDD search listed two domain hits, one of vWFA super family (cl00057) and the other of YfbK (COG2304). The top five BLAST hits of Q6KZG4 and Q6KZB9 indicated that these proteins contained a DUF929 domain and were prevalent in *Picrophilus* spp. and *Ferroplasma* spp. InterPro, Pfam and CDD indicated presence of domains of undetermined function. The top five BLAST hits of Q6L1G3 indicated that it was

**Table 1.  Details of the 30 secretory proteins identified in the secretome of *P. torridus* by LC MS and the signal peptides predicted by various signal peptide predictors.**

| S. No. | Protein accession number | Protein name/function | Gene name |
|---|---|---|---|
| 1 | Q6L2C5 | Hypothetical membrane associated protein | PTO0292 |
| 2 | Q6KZS2 | Thermosome subunit/protein folding | PTO1195 |
| 3 | Q6L182 | Oligopeptide ABC transporter Opp1/transmenbrane protein | PTO0685 |
| 4 | Q6KZF2 | Glutamate dehydrogenase/aminoacid metabolism | PTO1315 |
| 5 | Q6L2N6 | Extracellular solute-binding protein/membrane protein | PTO0181 |
| 6 | Q6L2M0 | Quinoprotein dehydrogenase/membrane protein | PTO0197 |
| 7 | Q6L202 | Elongation factor 1-alpha (EF-1-alpha) (Elongation factor Tu) (EF-Tu)/Protein biosynthesis | PTO0415 |
| 8 | Q6L0B7 | 2-oxoglutarate synthase, alpha chain (EC 1.2.7.3) | PTO1000 |
| 9 | Q6L0Y1 | Oligosaccharyl transferase STT3 subunit | PTO0786 |
| 10 | Q6KZA7 | Pyruvate ferredoxin oxidoreductase, alpha chain/pyruvate synthesis | PTO1360 |
| 11 | Q6L2C8 | Uncharacterized protein | PTO0289 |
| 12 | Q6L248 | Glutaredoxin related protein/electron transfer | PTO0369 |
| 13 | Q6L0C3 | Malate dehydrogenase/carbohydrate metabolism | PTO0994 |
| 14 | Q6KZG4 | Hypothetical exported protein | PTO1303 |
| 15 | Q6L140 | Peroxiredoxin 2/peroxidase activity | PTO0727 |
| 16 | Q6L2N0 | Membrane associated serine protease | PTO0187 |
| 17 | Q6KZB9 | Hypothetical membrane associated protein | PTO1348 |
| 18 | Q6L1G3 | Hypothetical exported protein | PTO0604 |
| 19 | Q6KZE9 | Iron(III) dicitrate ABC transporter extracellular binding protein/integral component of membrane | PTO1318 |
| 20 | Q6L1T2 | D-gluconate/D-galactonate dehydratase/D-gluconate catabolic process | PTO0485 |
| 21 | Q6KZT9 | ABC transporter extracellular solute-binding protein/membrane component | PTO1178 |
| 22 | Q6L2S5 | Uncharacterized protein | PTO0142 |
| 23 | Q6L2L9 | Uncharacterized protein | PTO0198 |
| 24 | Q6L268 | Hypothetical membrane protein | PTO0349 |
| 25 | Q6L1Y4 | Uncharacterized protein | PTO0433 |
| 26 | Q6L0M9 | CBS domain containing protein | PTO0888 |
| 27 | Q6L081 | Sugar ABC transporter 1/extracellular binding protein | PTO1036 |
| 28 | Q6KZK5 | Uncharacterized protein | PTO1262 |
| 29 | Q6L0W3 | Proteasome subunit alpha (Proteasome core protein)/protein degradation | PTO0804 |
| 30 | Q6L1B1 | 50S ribosomal protein L6/translation | PTO0656 |

a hypothetical, exported protein in *Picrophilus* spp., *Thermoplasma* spp. and Thermoplasmatales archaeon A-plasma. InterPro, Pfam and CDD revealed presence of a cell adhesion related domain found in bacteria (CARDB). The top five hits of Q6L2S5 indicated it to be a hypothetical protein present in *Picrophilus* spp., and *Thermoplasma* spp. InterPro and CDD indicated the presence of a domain of DrsEFH/DsrE superfamily while pfam failed to identify any domain. The top five hits of Q6L2L9 indicated it to be a transcriptional regulator of ArsR family present in archaeal organisms like *Acidiplasma* spp and *Ferroplasma* spp. InterPro identified a DNA-binding domain and CDD revealed the presence of a domain of COG4738 super family (accession: cl01956) which might function as a transcriptional regulator. Pfam did not suggest any domain. The top five hits of Q6L268 and Q6KZ5 indicated that these were hypothetical proteins of *Picrophilus* spp., and *Ferroplasma* spp. InterPro, Pfam and CDD did not reveal any conserved domains in these proteins. The top five hits of Q6L1Y4 indicated that it was a transcriptional regulator found in *Picrophilus* and *Thermoplasm*a spp. Pfam did not identify any domain in Q6L1Y4t but InterPro revealed a DNA-binding domain and CDD revealed the presence of a domain of phenylalanyl-tRNA synthetase subunit alpha.

**Table 2. Information about domains in the hypothetical/uncharacterized proteins in *P. torridus* secretome discerned using InterPro 74, Conserved Domain Database (CDD) and Pfam 32.**

| S. No. | Protein accession number | Gene name | InterPro 74 | Conserved Domain Database | Pfam 32 |
|---|---|---|---|---|---|
| 1 | Q6L2C5 | PTO0292 | Protein of unknown function DUF929 (IPR009272) | Reo_sigmaC superfamily | No result |
| 2 | Q6L2C8 | PTO0289 | von Willebrand factor, type A (IPR002035), archaellum regulatory network B, C-terminal (IPR040929) | vWFA superfamily (cl00057), YfbK (COG2304) | von Willebrand factor type A domain (PF00092), archaellum regulatory network B, C-terminal (PF18677) |
| 3 | Q6KZG4 | PTO1303 | IPR009272 (protein of unknown function DUF929) | DUF929 (pfam06053) | Domain of unknown function (PF06053) |
| 4 | Q6KZB9 | PTO1348 | No result | DUF929 (pfam06053) | Domain of unknown function (PF06053) |
| 5 | Q6L1G3 | PTO0604 | CARDB domain (IPR011635), Ig-like_fold (IPR013783) | CARDB superfamily (cl22904) | CARDB (PF07705) |
| 6 | Q6L2S5 | PTO0142 | DsrEFH-like (IPR027396) | DrsE superfamily (cl00672) | No result |
| 7 | Q6L2L9 | PTO0198 | Uncharacterized conserved protein UCP037373, transcriptional regulator, AF0674 (IPR017185), Winged helix-like DNA-binding domain superfamily (IPR036388) | COG4738 superfamily (cl01956) | No result |
| 8 | Q6L268 | PTO0349 | No result | No result | No result |
| 9 | Q6L1Y4 | PTO0433 | Winged helix-like DNA-binding domain superfamily (IPR036388) | pheS superfamily (cl30524) | No result |
| 10 | Q6KZK5 | PTO1262 | No result | No result | No result |

## Functional analysis of the secreted proteins

The secreted proteins were assigned functional categories according to the annotation derived from the *P. torridus* genome sequence (NCBI Reference sequence: NC_005877.1). It was observed that majority of the secreted proteins were membrane proteins, followed by proteins involved in other activities, followed by proteins involved in oxidoreductase activities, proteins involved in ion binding activity, peptidase activity, structural constituents of ribosome, GTPase activity, rRNA binding, peptidase, lyase and transferase activities (Fig 1). Analysis of functional enrichment by Gene Ontology (GO) revealed that the secreted proteins were involved in a variety of biological processes (BP) of which the major function was metabolic processes (Fig 2A). In the category molecular function (MF), the secreted proteins were involved in oxidoreductase activity (6 proteins), ion binding (3 proteins) and peptidase activity (2 proteins). One protein each was found to be involved in RNA binding, rRNA binding, structural constituent of ribosome, ligase activity, translation factor activity, DNA binding, GTPase activity, unfolded protein binding (Fig 2B). Cell component (CC) enrichment analysis revealed that most of the secretory proteins were cytoplasmic proteins (6 proteins), followed by intracellular, ribosomal, cell and macromolecular complex proteins (1 protein each) (Fig 2C).

## Protein-protein interactions (PPIs) and KEGG pathway analysis

STRING analysis revealed that except one protein (Q6KZG4), all the proteins had known or predicted interacting partners. According to the STRING database, the 29 secretory proteins of *P. torridus* interacted with 488 other proteins of *P. torridus*. KEGG pathway map of the 30 proteins revealed their involvement in 29 different pathways (**Table 3**). The interaction network of the secretory proteins was created using Cytoscape (Fig 3). The secretory proteins are marked inside the squares, their interacting proteins are marked in circles and their respective

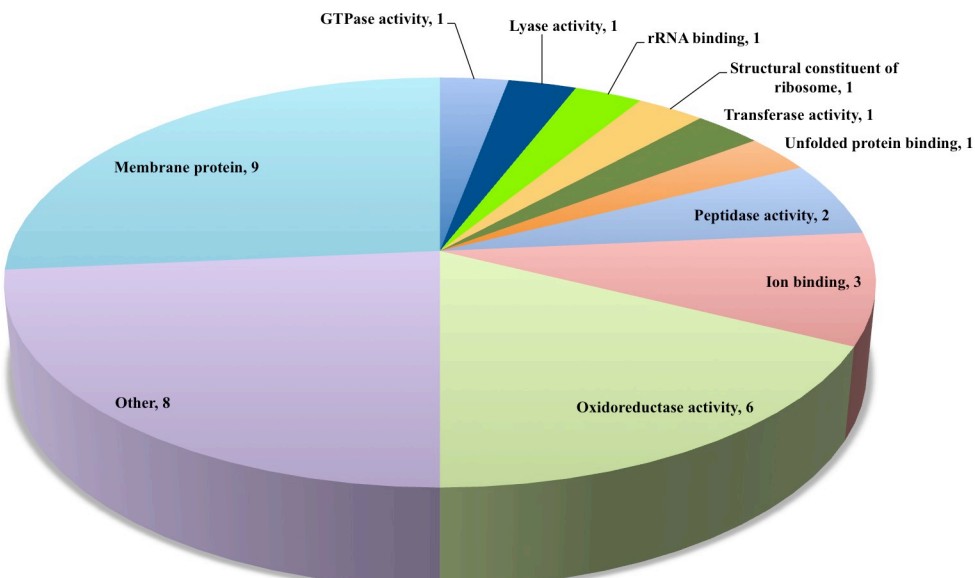

**Fig 1. Distribution of *P. torridus* secretory proteins according to their functional categories.**

pathways are depicted via a particular colour. Since, Cytoscape can show a single pathway at a time, hence only a single pathway has been depicted for some multi-functional protein(s).

## Prediction efficacy of SPPs

The average prediction efficacy of PRED-SIGNAL and PRED-TAT in identifying SPs in the secretory proteins of *P. torridus* identified in three independent experiments (S1–S3 Files)

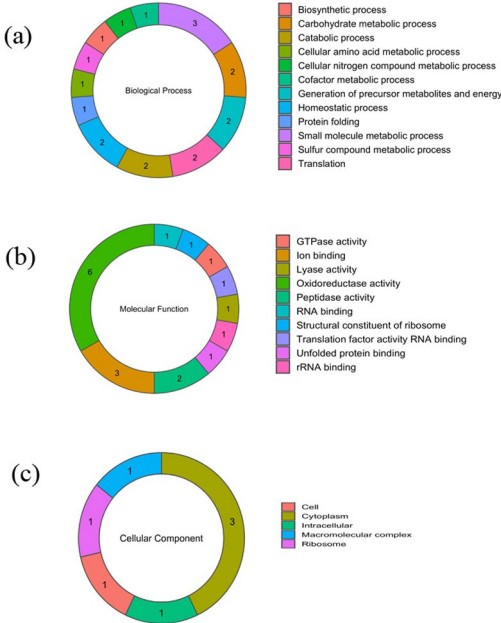

**Fig 2.** Functional categories of *P. torridus* secretory proteins on the basis of Gene Ontology (GO): (a) biological function (b) molecular function and (c) cellular component function.

**Table 3. Details of *P. torridus* secretory proteins involved in the various KEGG pathways.**

| S.No. | KEGG pathway | Protein accession number |
|---|---|---|
| 1. | Carbon metabolism | Q6L1T2, Q6L0C3, Q6L0B7, Q6KZF2, Q6KZA7 |
| 2. | Microbial metabolism in diverse environments | Q6L1T2, Q6L0C3, Q6L0B7, Q6KZF2, Q6KZA7 |
| 3. | Biosynthesis of antibiotics | Q6L0C3, Q6L0B7, Q6KZA7 |
| 4. | Biosynthesis of secondary metabolites | Q6L0C3, Q6L0B7, Q6KZA7 |
| 5. | Pyruvate metabolism | Q6L0C3, Q6L0B7, Q6KZA7 |
| 6. | Citrate cycle | Q6L0C3, Q6L0B7, Q6KZA7 |
| 7. | Carbon fixation pathways in prokaryotes | Q6L0C3, Q6L0B7, Q6KZA7 |
| 8. | Butanoate metabolism | Q6L0B7, Q6KZA7 |
| 9. | Glycolysis/Gluconeogenesis | Q6L0B7, Q6KZA7 |
| 10. | Galactose metabolism | Q6L1T2 |
| 11. | Cysteine and methionine metabolism | Q6L0C3 |
| 12. | Pentose phosphate pathway | Q6L1T2 |
| 13. | D-Glutamine and D-glutamate metabolism | Q6KZF2 |
| 14. | ABC transporters | Q6L081 |
| 15. | Glyoxylate and dicarboxylate metabolism | Q6L0C3 |
| 16. | Proteasome | Q6L0W3 |
| 17. | Arginine biosynthesis | Q6KZF2 |
| 18. | Alanine aspartate and glutamate metabolism | Q6KZF2 |
| 19. | Ribosome | Q6L1B1 |
| 20. | Nitrogen metabolism | Q6KZF2 |
| 21. | Methane metabolism | Q6L0C3 |

were almost similar (~16%), followed by Signal P (15.07%) and LipoP (13.55%) (S2 Table). Evaluation of the prediction efficacy of SPPs in identifying SPs in the 30 proteins that were common in three independent secretome samples revealed that, each SPP identified N-terminal signal sequences in eight different proteins of *P. torridus* (Table 4). Thus, the prediction efficacy of each SPP was 26.66%. However, all the four SPPs identified N-terminal SPs in five proteins of *P. torridus* namely, Q6L2C5, Q6L182, Q6L0C3, Q6L1G3 and Q6L081. PRE-D-SIGNAL, PRED-TAT and LipoP identified SPs in the protein Q6KZB, while SignalP, PRE-D-TAT and LipoP identified SPs in protein Q6KZG4. Both PRED-SIGNAL and SignalP identified SPs in *P. torridus* proteins Q6L2N0 and Q6KZE9. Both PRED-SIGNAL and SignalP made identical predictions, except for the protein Q6KZG4 in which SP was predicted by SignalP, while PRED-SIGNAL identified trans membrane segments in this protein. PRE-D-SIGNAL identified the SPs in protein QCKZB9, but SignalP could not. Though, predictions by PRED-TAT and SignalP were similar, PRED-TAT additionally identified SPs in the protein Q6L2S5. Like SignalP and LipoP, PRED-TAT also identified SPs in the protein Q6KZG4. PRE-D-TAT identified transmembrane segments in Q6L2N0 and Q6KZE9, while PRED-SIGNAL and SignalP predicted SPs in these proteins. LipoP and PRED-TAT identified SPs in the proteins Q6KZT9 and Q6L2S5, respectively. Though, most of the LipoP predictions were similar to other predictors, unlike PRED-SIGNAL and SignalP, it identified SPs in proteins Q6KZG4 and Q6KZT9.

## Discussion

The aim of the current study was to evaluate the efficacy of various SPPs in identifying SPs in the experimental secretome of *P. torridus*. Culture filtrate proteins of *P. torridus* were concentrated, processed and analyzed by LC MS. Using this approach; 68 proteins were identified by

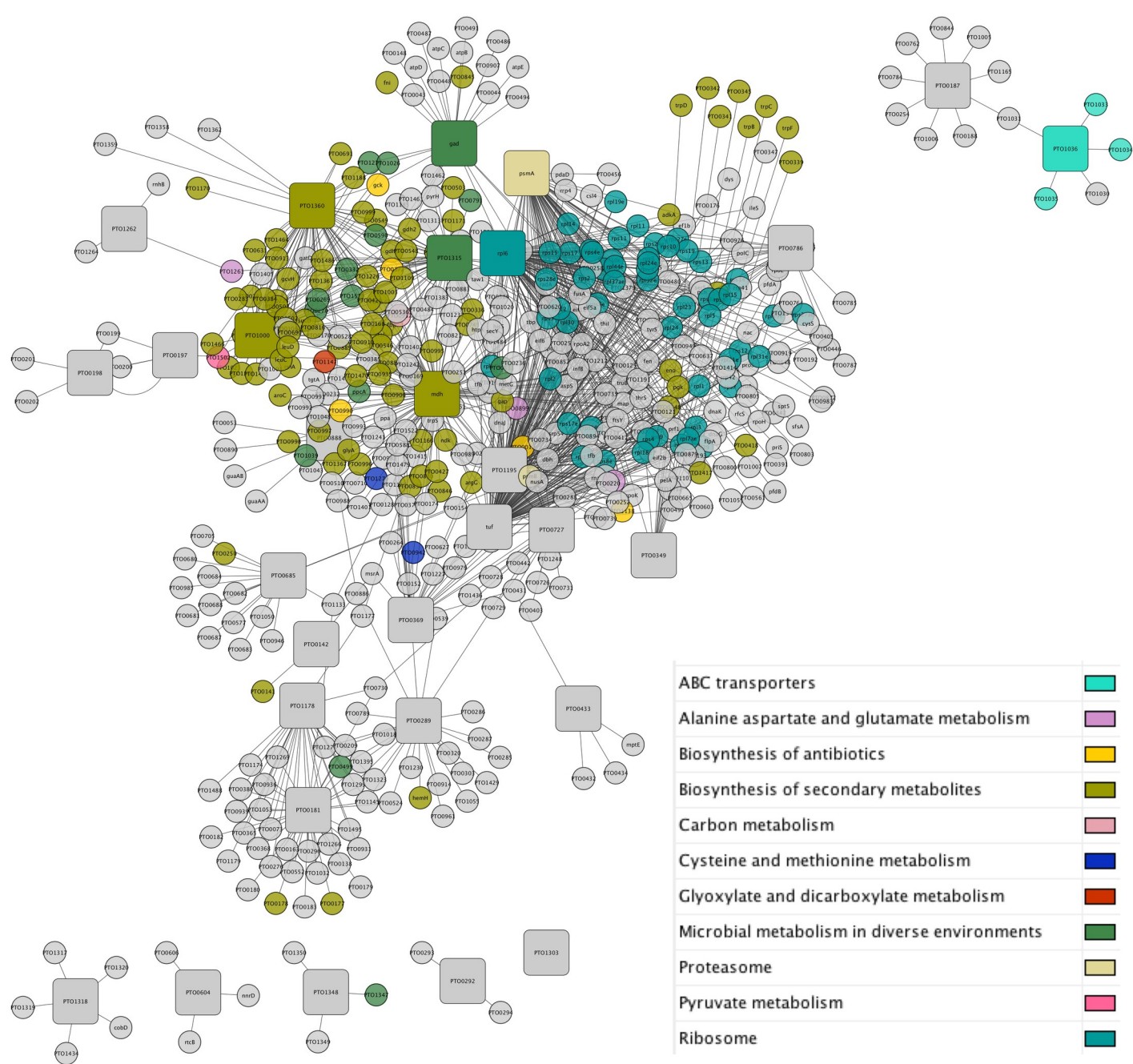

**Fig 3. The protein-protein interaction network and KEGG pathway map of the secretory proteins of *P. torridus*.**

LC MS in the first experiment (S1 File), 75 proteins in the second experiment (S2 File) and 97 proteins in the third experiment (S3 File). To avoid any ambiguity and remove any technical artefacts, only 30 proteins which were present in all the three experiments were included in this study. In depth analysis of the experimental secretome of *P. torridus* revealed that majority of the secreted proteins were involved in various metabolic processes and one-third of the secreted proteins were hypothetical/uncharacterized. In this regard, our results are similar to an earlier study which reported that most of the annotated secreted proteins of *P. torridus*

**Table 4. Details of the signal peptides, their cleavage sites and trans membrane segments predicted by various signal peptide predictors in the secretory proteins of *P. torridus*.**

| S. No. | Protein accession number | Signal peptide predictor (signal peptides and signal cleavage site) | | | |
|---|---|---|---|---|---|
| | | PRED-SIGNAL | SignalP 5.0 | PRED-TAT | LipoP 1.0 |
| 1 | Q6L2C5 | MDNKKIISIAMVAIMVLSAFAVLGSMPV QQAATHNKA signal cleavage site (37–38) | MDNKKIISIAMVAIMVLSAFAVLGSM PVQQA signal cleavage site (31–32) | MDNKKIISIAMVAIMVLSAAVLGS MPVQQA signal cleavage site (31–32) | MDNKKIISIAMVAIMVLSAFAVLG SMPVQQA, signal cleavage site (31–32) |
| 2 | Q6L182 | MSESDYRKKFKFKYMLIAAVLIVSLIFVA EGFGAAIPGQTSAPAVA signal cleavage site (45–46) | MSESDYRKKFKFKYMLIAAVLIVSLIFVA EGFGAA signal cleavage site (34–35) | MSESDYRKKFKFKYMLIAAVLIVSLIFVAEGFGAAIP GQTSAPAVA signal cleavage site (45–46) | MSESDYRKKFKFKYMLIAAVLIVSLIF VAEGFGA signal cleavage site (33–34) |
| 3 | Q6L2N6 | TM (8–28) | - | TM (8–28) | - |
| 4 | Q6L2M0 | TM (7–26) | - | TM (10–30) | - |
| 5 | Q6L0Y1 | - | - | TM (53–73) | - |
| 6 | Q6L0C3 | MARSKISVIGAGAVGATVAQTLA signal cleavage site (23–24) | MARSKISVIGAGAVGATVAQTLA signal cleavage site (23–24) | MARSKISVIGAGAVGATVAQTLAIR signal cleavage site (25–26) | MARSKISVIGAGAVGATVA signal cleavage site (19–20) |
| 7 | Q6KZG4 | MANINYKLLVLFIAVFVVIAFFAVDYDLYHA signal cleavage site (31–32) | MANINYKLLVLFIAVFVVIAFFAVDYDLYHA signal cleavage site (31–32) | MANINYKLLVLFIAVFVVIAFFA signal cleavage site (23–24) | MANINYKLLVLFIAVFVVIAFFA signal cleavage site (23–24) |
| 8 | Q6L2N0 | MRGIKIIAIIIICMFIITS signal cleavage site (19–20) | MRGIKIIAIIIICMFIITSMDVVIP signal cleavage site (25–26) | TM (4–24) | - |
| 9 | Q6KZB9 | MAKNNKRSTNKNQKNKNSASKNQNKKN NINLKNKNVIGSAIAAVIIVVLVVVVLTHPLYR signal cleavage site (64–65) | - | MAKNNKRSTNKNQKNKNSASKNQNKKNNIN LKNKNVIGSAIAAVIIVVLVVVVLTHPLYR signal cleavage site (60–61) | MAKNNKRSTNKNQKNKNSASKNQNKKNNINL KNKNVIGSAIAAVIIVVLVVVVLT signal cleavage site (55–56) |
| 10 | Q6L1G3 | MNKTRRGIIVAVTLLMVLSTFAFVSQA signal cleavage site (27–28) | MNKTRRGIIVAVTLLMVLSTFAFVSQA signal cleavage site (27–28) | MNKTRRGIIVAVTLLMVLSTFAFVSQA signal cleavage site (27–28) | MNKTRRGIIVAVTLLMVLSTFAFVSQA signal cleavage site (27–28) |
| 11 | Q6KZE9 | MNKKVIASLIIVVIIIISGISYYIHSNTATSGKITVKA signal cleavage site (39–40) | MNKKVIASLIIVVIIIISGISYYYIHS signal cleavage site (27–28) | TM (5–25) | - |
| 12 | Q6KZT9 | TM (7–29) | - | TM (10–30) | MVMNSKARIIIAVVVVIIIIAAGFSFA signal cleavage site (27–28) |
| 13 | Q6L2S5 | - | - | MKNVAIIISTSNKEKAVA signal cleavage site (18–19) | - |
| 14 | Q6L268 | TM (35–63) | - | TM (36–66) | - |
| 15 | Q6L081 | MAKNKIIAIVAIVIVIIVIGSVIA signal cleavage site (24–25) | MAKNKIIAIV AIVIVIIVIGSVIA signal cleavage site (24–25) | MAKNKIIAIVAIVIVIIVIGSVIA signal cleavage site (24–25) | MAKNKIIAIVAIVIVIIVIGSVIA, signal cleavage site (24–25) |

TM: Trans Membrane segment.

Number in parenthesis indicates amino acid position.

were components of the respiratory chain or hypothetical proteins, transporters, proteases and exported binding proteins [2]. Despite the fact that many of these proteins had intracellular functions, and should not be present in culture filtrate, intracellular proteins have been regularly reported from culture filtrates of archaea [14,29] and bacteria [30]. If this is due to the artefacts during cell lysis or due to active secretion of intracellular proteins in the surrounding culture medium [31], the underlying reason is still unclear. In archaea, protein export via membrane vesicles has been proposed as another possible reason underlying the presence of these proteins in culture filtrate [32,33]. An earlier study reported that various proteins involved in translation and, energy and metabolism were exported via secreted membrane vesicles in archaeal *Sulfolobus* species [14]. Interestingly, in our study too, some secreted proteins like malate dehydrogenase (Q6L0C3) were observed to be involved in many different metabolic pathways like carbon metabolism, microbial metabolism in diverse environments, biosynthesis of antibiotics, biosynthesis of secondary metabolites, pyruvate metabolism, citrate cycle etc. Multifunctional secreted proteins might be an important attribute for thermoacidophilic adaptation of *P. torridus* which has the smallest genome (1.5 Mbp) among nonparasitic aerobic microbes.

Of the 30 proteins discerned in the experimental secretome, ten proteins were identified as hypothetical/uncharacterized proteins. Due to absence of any conserved domain(s) or domain(s) of underdetermined functions, putative functions of four secreted hypothetical proteins— Q6KZG4, Q6L268, and Q6KZK5 and Q6KZB9 could not be predicted *in silico*. Of the six hypothetical secreted proteins whose putative functions could be predicted, two proteins, Q6L2L9 and Q6L1Y4 were probably transcriptional regulators. Proteins containing domains of Reo_sigmaC super family have been reportedly involved in host-virus interactions, hence it might be anticipated that Q6L2C5 might also be involved in *Picrophilus*-viral interactions [34]. The secreted protein Q6L2C8 contained a Von Willebrand factor type A (vWA) domain which was originally found in the blood coagulation protein von Willebrand factor (vWF) where it helps in the formation of protein aggregates [35]. The vWA domain containing proteins are involved in a variety of important cellular functions like formation of the basal membrane formation, signalling, cell migration, cell differentiation, adhesion, haemostasis, chromosomal stability and in immune defences. Thus, it might be anticipated that Q6L2C8 might also be involved in vital cellular functions of *P. torridus*. Interestingly proteins containing a vWA domain have been reported to be present in secreted membrane vesicles of archaeal *Sulfolobus* species [14]. The protein Q6L1G3 contained a domain related to cell adhesion in bacteria (CARDB). Proteins containing CARDB domain were reported to be homologs of calpain which is an essential, cytoplasmic, calcium-dependent cysteine endopeptidase of eukaryotes [36]. Calpains are implicated in a variety of calcium-regulated cellular processes in eukaryotes such as signal transduction, cell proliferation, cell cycle progression, differentiation, apoptosis, *etc* [37,38]. Thus, Q6L1G3 might also be involved in various calcium-regulated cellular processes of *P. torridus*. The protein Q6L2S5 contained a DsrF-like family domain. DsrE/DsrF are small soluble proteins which are involved in intracellular reduction of sulphur [39]. Hence, the protein Q6L2S5 might help in survival of *P. torridus* in solfataric environment.

The prediction efficacy of the four SPPs on the 30 proteins which were common in the three independent secretome samples was identical (26.66%) because each program identified SPs in eight different proteins of *P. torridus*. The supplementary information contained in the SP prediction program PRED-SIGNAL showed that 86 proteins of *P. torridus* have SPs, while *in silico* predictions by SignalP revealed that 121 proteins of *P. torridus* were secretory proteins [2]. Till recently, PRED-SIGNAL was the only program available for prediction of archaeal SPs. Since, it was trained on archaeal secretome, its prediction accuracy was expected to be better than other prediction programs. However, our results revealed that it could identify SPs in

only eight proteins, and trans membrane segments in five proteins. Though the earlier versions of SignalP could predict SPs in secretory proteins gram-positive and–negative bacteria, the latest version, SignalP 5.0 can predict the SPs in archaeal proteins, also [40,41]. However, our results revealed that SignalP 5.0 could also identify SPS in only eight proteins of *P. torridus*. This suggests that the experimental secretome of *P. torridus* might be smaller than the theoretical secretome predicted by various SPPs. However, there might be several reasons underlying the differences observed in the experimental and theoretical secretome. Of which, the first might be that, the SignalP was trained on SPs of gram-positive and- negative bacteria which might have led to an over estimated number of SPs in *P. torridus*, which is an archaea. Second, secretome profile of microorganisms varies greatly in accordance with their growth conditions and different stages of growth (log phase versus exponential phase). Thus, the *P. torridus* secretome reported in the present study might be specific to the growth conditions which were used in this study. Third, some low- level expressed proteins might have been missed in this study from proteomic identification, due to technical constraints like, detection limit of mass spectrometry.

Since TAT substrates have been reportedly present in the secretome of archaea [13] the SPP PRED-TAT was used to identify TAT substrates in the secretome of *P. torridus*. PRED-TAT predicts twin-arginine and secretory signal peptides using Hidden Markov Models [18]. Of the 30 proteins, PRED-TAT identified SPs in eight proteins and trans membrane segments in seven proteins. PRED-TAT did not identify any TAT substrates in the secretome of *P. torridus*. Lipoproteins were also reportedly abundant in the secretome of archaea [5] hence; their presence in the experimental secretome of *P. torridus* was investigated using the SPP LipoP. Though, lipoproteins are usually attached to the cell membrane, they might also be present in the culture filtrate due to natural shedding [42,43]. LipoP predicted that eight proteins harbored SPs, were transported via the standard Sec/SPI pathway and none of them was a lipoprotein.

The predictions by PRED-TAT and LipoP suggest that TAT substrates and lipoproteins might be absent in the secretome of *P. torridus*. Also, the fact that N-terminal SPs were identified in only a small fraction of the experimental secretome of *P. torridus* suggests two plausible underlying reasons. Either, the SPPs used in this study were less efficient in identifying archaeal SPs or protein transloction in *P. torridus* does not take place only *via* the general SP-dependent, Sec-pathway. Additionally, there might be alternative mechanisms of protein transport in *P. torridus* like, secreted membrane vesicles as reported earlier in archaeal *Sulfolobus* species [14].

## Conclusion

The information about secreted proteins of archaea is still fragmentary. The present study adds to the slowly growing knowledge base of archaeal secretomes and is the first study about secretome of *P. torridus*. Under the specific growth conditions which were used in this study, 30 proteins of *P. torridus* were identified as secreted proteins by LC MS. TAT substrates frequently reported from the secretome of haloarchaea [13] and lipoproteins reportedly abundant in the secretome of *P. furious* [5] were found to be completely absent in the secretome of *P. torridus*. The majority of the secreted proteins were predicted to be involved in metabolic pathways. Since, vWA domain containing proteins, were reportedly exported via secreted membrane vesicles in archaeal *Sulfolobus* species [14] hence, it can be speculated that the hypothetical protein of *P. torridus* with such domains might also be exported by membrane vesicles. The four SPPs used in this study, PRED-SIGNAL, SignalP, PRED-TAT and LipoP identified N-terminal SPs in a small fraction of the secreted proteins. This indicates that either

these four SPPs were insufficient in identifying the N-terminal signal sequences or N-terminal signal sequences might not exist in majority of the secreted proteins of *P. torridus*. This suggests that there might be alternative mechanisms of protein translocation in *P. torridus* like, secretory membrane vesicles, as reported for *Sulfolobus* spp [14]. However, further experiments are required to corroborate our findings. Nevertheless, this preliminary study is expected to provide a useful basis for further studies on protein translocation in this thermoacidophilic archaeon.

## Supporting information

**S1 Fig. Circos plot showing the domains present in the 30 secretory proteins of *P. torridus*.**
(PNG)

**S1 Table. *P. torridus* secretory proteins, PDB BLAST hits with percentage identity.**
(DOCX)

**S2 Table. Prediction efficacy of various SPPs in identifying SPs in *P. torridus* secretome identified in three independent experiments.**
(DOCX)

**S1 File. *P. torridus* proteins identified by LC MS in the first experiment.**
(XLSX)

**S2 File. *P. torridus* proteins identified by LC MS in the second experiment.**
(XLSX)

**S3 File. *P. torridus* proteins identified by LC MS in the third experiment.**
(XLSX)

## Author Contributions

**Conceptualization:** Manisha Goel.

**Data curation:** Neelja Singhal, Anjali Garg, Nirpendra Singh, Manish Kumar.

**Formal analysis:** Neelja Singhal, Nirpendra Singh, Manish Kumar.

**Funding acquisition:** Manisha Goel.

**Investigation:** Neelja Singhal, Anjali Garg, Nirpendra Singh, Pallavi Gulati, Manish Kumar.

**Methodology:** Neelja Singhal, Pallavi Gulati.

**Project administration:** Manisha Goel.

**Resources:** Manisha Goel.

**Software:** Anjali Garg.

**Writing – original draft:** Neelja Singhal.

**Writing – review & editing:** Manisha Goel.

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
