## [Decision Letter · Decision Letter 0]

20 Apr 2021

PONE-D-21-06360

Efficacy of signal peptide predictors in identifying signal peptides in the experimental secretome of Picrophilous torridus, a thermoacidophilic archaeon

PLOS ONE

Dear Dr. Goel,

Thank you for submitting your manuscript to PLOS ONE. After careful consideration, we feel that it has merit but does not fully meet PLOS ONE’s publication criteria as it currently stands. Therefore, we invite you to submit a revised version of the manuscript that addresses the points raised during the review process.

We look forward to receiving your revised manuscript.

Kind regards,

Dinesh Gupta

Academic Editor

PLOS ONE

Journal Requirements:

Reviewers' comments:

Reviewer's Responses to Questions

**Comments to the Author**

1. Is the manuscript technically sound, and do the data support the conclusions?

Reviewer #1: Yes

Reviewer #2: Partly

2. Has the statistical analysis been performed appropriately and rigorously? 

Reviewer #1: N/A

Reviewer #2: Yes

3. Have the authors made all data underlying the findings in their manuscript fully available?

Reviewer #1: Yes

Reviewer #2: Yes

4. Is the manuscript presented in an intelligible fashion and written in standard English?

Reviewer #1: Yes

Reviewer #2: Yes

5. Review Comments to the Author

Reviewer #1: [Major comment]

1. It’s not clear why author predicted the N-terminal sequences and what is the take-home message from that analysis.

2.Results are poorly represented as the Figures. I recommend authors to show proteins domains in the 30 secretome proteins (e.g., stacked over each other) for better visualization, similar to how HMMER/Pfam/Interpro represent. Authors may also use CIRCOS plots to highlight functional domains of 30 proteins as a condensed representation.

3.Similarly, predicted N-terminal regions of interest may also be visualized.

4.Please provide phylogenetic comparison of secretome of P. torridus with known secretome of related archaea.

5.Is there any secreted protein (out of 30) for which 3D structure is available or can be predicted using current methods? A result Table with PDB-blast may help here.

6.Are there any effector proteins predicted?

[Minor comments]

1.Line 98: The text describing the tools is not required in “introduction” section. Please move the text to the “method” section.

2.Line 142 and 174: Please explain how UniProtKB ID was assigned to filtrate proteins. Please cite UniProtKB.

3.Line 147: For STRING database, what parameters and sources were used for shortlisting interacting proteins.

4.Line 165: The section “Analysis of N-terminal signal sequences”, needs more details. It’s not clear what authors mean by “analysis” here.

5.Please use first letter in capital, whenever abbreviations are used, e.g., in Table 1 please use Trans Membrane.

6.Table 2. It should be - Protein accession number. Please use proper sentence case in other part of the manuscript too.

7.Line 232: Please use small paragraph heading. I think, name of the tools in heading is not required.

8.Line 237: Please do not write version number of interpro or pfam everywhere in the entire manuscript. Its sufficient to cite version number in method section only.

9.Line 353: Please remove the result hyperlink for PRED-SIGNAL.

10.Line 345: The section “Prediction efficacy of various SP predictors for the experimental secretome of P. torridus” should be in the result section (within the section “Identification of N-terminal signal sequences”), instead of discussion. Also results of this section must be presented as a separate Table or figure, as its too confusing to read and summarize the different variables.

11.A review from native English speaker may be required to fix minor grammatical errors.

Reviewer #2: Authors generated the LC MS data of P. torridus and predicted the efficacy of signal peptide prediction software. Thought the work is of interest, following points should be addressed:

# Authors should highlight 1-2 metabolic pathways in the abstract.

# Authors conclude 3 possibilities in conclusion:

“currently available SP prediction programs” SPP are insufficient / inefficient?

N-ter SP absent in P. t.

Alternative mech. of protein translocation in P. t.

So “currently available SP prediction (SPP) programs” -- > “these 4 SP prediction programs” should be used to tone down the conclusion.

Authors should include the latest methods e.g. How many software are available. Why did not use latest ones e.g. DeepSig

What is rationale of using these 4 software.

# Please make suppl. table or excel showing how many SPPs are based on archaea datasets to identify archaea SPs. Provide the very briefly basic details of the methods along with the data sample size especially archaea data used for machine learning model development.

# Please provide some quantitative manner to show the “Efficacy of signal peptide predictors in identifying signal peptides”, e.g. %age of proteins found to have SP using 4 software.

Additionally, authors should provide for all full 3 sets in supplementary file the prediction score of all 4 software.

# Authors should order these SPPs based on some rationale. E.g. year of publish or performance? and be consistent in describing these methods all over in that order only.

# on line 353, this thing is counter intuitive.

"The supplementary information enlisted in PRED-SIGNAL indicated that 86 proteins of P. torridus were secretory proteins (http://bioinformatics.biol.uoa.gr/PRED-SIGNAL-results/). However, when the 30 experimentally derived secreted proteins were submitted to PRED-SIGNAL it identified signal peptides in only eight proteins, while trans membrane segments were identified in five proteins."

Were your 30 sequences among 1535 seq mentioned on that page of signal-pep?

If yes, then how only 8 are shown to have SPs?

What input was given when you say "when the 30 experimentally derived secreted proteins were submitted to PRED-SIGNAL"?

Was the sequence of your 30 proteins and their 30 out of 1535 proteins different or identical?

Authors need to explain this section.

# line 80, soluble in what? Water or lipid?

# For Ref 17, what data was used to make prediction model. It is based on archaeal protein.?

# Line 116, is there any reference for such protocol used in this section, authors may cite that.

# protein pilot software is not mentioned in the text while suppl. table 1 mentions it.

# Why CPU time and rates are missing for other rows in excel sheet Speed and Distribution Analysis in suppl file 1, sheet 1

# Why Global FDR is recommended at 5 and 10% unlike Global FDR fit at 1%? In protein FDR Summary sheet in supp. Tables

# Authors should mention version of all software and databases used in the study. Which version of backend database was used for blast.

# What is meaning of % Coverage (95) and Peptides (95%) in table 1 should be mentioned.

# Authors should sort table 1 based on some criteria, so that it is easy for readers to comprehend.

# Line 178: Pfam 32.0 failed to predict any domain in the hypothetical protein Q6L265? Did not find any protein with this name Q6L265. Also, what is the reason for not finding any domain?

# Table 1 is least described in the text. Expand it like authors explain table 2. Table 1 is explained rather in discussion.

# Table 1 contains 30 proteins while fig 1 contains 34?

# Order fig 1 based on counts.

# Fig 2, reorder based on counts. Also, the legend title is ‘Function’ for all the panels, it should be corrected.

# line 226 "Here, due to the limitations of the Cytoscape software tool, we have shown only one pathway of multi-functional protein(s)". It is not clear.

# Heading in lines 232/233 seem not in continuation. Please double check.

# Fig 3, Text mentions protein id while figure contains gene ids, need to be consistent for proper interpretation of the figure.

Mention about the size of nodes, length/thickness of edges if they mean anything or not.

What grey color represents in fig 3.

# Line 237. Be consistent in metnioning version name e.g. InterPro 74 and Pfam 32

# line 238 : "InterPro 74 and Pfam 32 could not find any domain, while CDD search predicted presence of a domain of Reo_sigmaC super family". But table 1 shows interpro finds a domain?

# Section of text in 235-265 should also be available as an additional table or suppl table

# Suppl files 1,2,3 need to be explained properly as they contain multiple sub sheets. Need to explain at least one suppl file with the related content and interpretation.

# Line 297/298: “…proteins might have been missed in this study from proteomic identification, due to technical constraints like, detection limit of mass spectrometry.” What is that limit in this study’s experiment.?

# English needs to be improved including semantic and grammatical errors e.g.

in line 85 sp. Vs spp.

artifacts vs technical artefacts,

have read and approve -- > have read and approved

6. PLOS authors have the option to publish the peer review history of their article (what does this mean?). If published, this will include your full peer review and any attached files.

Reviewer #1: No

Reviewer #2: No

---

## [Author Response · Author response to Decision Letter 0]

7 Jul 2021

AUTHORS’ REPLY TO THE REVIEWERS’ COMMENTS

Reviewer #1: [Major comment]

Comment 1: It’s not clear why author predicted the N-terminal sequences and what is the take-home message from that analysis.

Authors’Reply: The aim of the present study was to identify the experimental secretome of P. torridus and evaluate the efficacy of efficacy of four signal peptide predictors (SPPs) - PRED-SIGNAL, SignalP 5.0, PRED-TAT and LipoP 1.0 in identifying SPs in the experimental secretome. It is important to study the microbial secretome because secretory proteins perform a variety of functions like degradation of complex polymeric substances (carbohydrates and proteins), passage of nutrients inside the cell, protection against toxic compounds, signal transduction etc and also helps in adaptation and survival in a particular niche, including thermoacidophilic environment. 

Comment 2: Results are poorly represented as the Figures. I recommend authors to show proteins domains in the 30 secretome proteins (e.g., stacked over each other) for better visualization, similar to how HMMER/Pfam/Interpro represent. Authors may also use CIRCOS plots to highlight functional domains of 30 proteins as a condensed representation.

Authors’ Reply: This point of the Reviewer is well taken. In the revised manuscript, a CIRCOS plot highlighting the functional domains of the 30 proteins as a condensed representation has been included. 

(please see page 7, line 187 of the revised manuscript and Supplementary Fig.1)

Comment 3: Similarly, predicted N-terminal regions of interest may also be visualized.

Authors’ Reply: This point of the Reviewer is well taken. The predicted N-terminal regions of the 30 proteins have been shown separately in Table 4.

Comment 4: Please provide phylogenetic comparison of secretome of P. torridus with known secretome of related archaea.

Authors’ Reply: To the best of our knowledge, secretomes of only five archaea have been discerned. These are - an antartic archaeon Methanococcoides burtonii, hyperthermoacidophilic archaeon Sulfolobus spp., hyperthermophilic archaeon Pyrococcus furiosus and haloarchaea like Haloferax volcanii and Natrinema sp. J7-2. It was not possible to perform phylogenetic comparison of the secretory proteins of P. torridus with the secretory proteins of these archaea because only a few proteins were common between them. 

Comment 5. Is there any secreted protein (out of 30) for which 3D structure is available or can be predicted using current methods? A result Table with PDB-blast may help here.

Authors’ Reply: This point of the Reviewer is well taken. A table showing the PDB-BLAST results of the secreted proteins has been included in the revised manuscript.

(please see page 7, line 187-190 of the revised manuscript and Supplementary Table 1 )

Comment 6: Are there any effector proteins predicted?

Authors’ Reply: None of the secretory protein of P. torridus was predicted as effector protein.

[Minor comments]

Comment 1: Line 98: The text describing the tools is not required in “introduction” section. Please move the text to the “method” section.

Authors’ Reply: This point of the Reviewer is well taken. The text describing the tools has been moved to the method section.

(please see page 6, line 168-177 of the revised manuscript)

Comment 2: Line 142 and 174: Please explain how UniProtKB ID was assigned to filtrate proteins. Please cite UniProtKB.

Authors’ Reply: The authors have deleted these lines from the revised manuscript because the UniProtKB ID of the secretory proteins were not assigned by us. These were given by the software during peptide identification by LC MS experiments.

Comment 3: Line 147: For STRING database, what parameters and sources were used for shortlisting interacting proteins.

Authors’ Reply: This point of Reviewer is well taken. The parameters and sources which were used for shortlisting interacting proteins with the STRING database have been included in the revised manuscript.

(please see page 6, lines 149-152 of the revised manuscript)

Comment 4: Line 165: The section “Analysis of N-terminal signal sequences”, needs more details. It’s not clear what authors mean by “analysis” here.

Authors’ Reply: The authors are thankful to the Reviewer for pointing this. In the revised manuscript the word “analysis” has been replaced with “identification”

Comment 5: Please use first letter in capital, whenever abbreviations are used, e.g., in Table 1 please use Trans Membrane.

Authors’ Reply: This point of Reviewer is well taken. The authors have checked and verified that the first letter in abbreviations is in capitals throughout the manuscript.

Comment 6: Table 2. It should be - Protein accession number. Please use proper sentence case in other part of the manuscript too.

Authors’ Reply: This point of Reviewer is well taken. The authors have replaced Uniprot accession number with Protein accession number and verified that proper sentence case has been used in other part of the manuscript

Comment 7: Line 232: Please use small paragraph heading. I think, name of the tools in heading is not required.

Authors’ Reply: This point of Reviewer is well taken. In the revised manuscript, small paragraph headings have been used and the name of the tools has been removed from the headings.

Comment 8 : Line 237: Please do not write version number of interpro or pfam everywhere in the entire manuscript. It is sufficient to cite version number in method section only.

Authors’ Reply: This point of Reviewer is well taken. In the revised manuscript, version number of interpro or pfam has been included only in the methods section. 

Comment 9 : Line 353: Please remove the result hyperlink for PRED-SIGNAL.

Authors’ Reply: This point of Reviewer is well taken. The result hyperlink for PRED-SIGNAL has been removed in the revised manuscript.

Comment 10 : Line 345: The section “Prediction efficacy of various SP predictors for the experimental secretome of P. torridus” should be in the result section (within the section “Identification of N-terminal signal sequences”), instead of discussion. Also results of this section must be presented as a separate Table or figure, as its too confusing to read and summarize the different variables.

Authors’ Reply: This point of the Reviewer is well taken. The section Prediction efficacy of various SP predictors for the experimental secretome of P. torridus has been moved to the results section and some of the lines have been re-written to enable easy understanding of the different variables. Also, the results of this section have been presented in a separate Table. 

(please see page 8-9, lines 261-281 of the revised manuscript and Table 4 )

Comment 11: A review from native English speaker may be required to fix minor grammatical errors.

Authors’ Reply: This point of the Reviewer is well taken. The authors have taken the help from a senior colleague to fix the minor grammatical errors.

Reviewer #2:

Authors generated the LC MS data of P. torridus and predicted the efficacy of signal peptide prediction software. Thought the work is of interest, following points should be addressed:

Comment 1: # Authors should highlight 1-2 metabolic pathways in the abstract.

Authors’Reply: This point of the Reviewer is well taken. A metabolic pathway has been highlighted in the abstract section.

(please see page 1, lines 47-51 of the revised manuscript).

Comment 2: # Authors conclude 3 possibilities in conclusion:

“currently available SP prediction programs” SPP are insufficient / inefficient?

N-ter SP absent in P. t. Alternative mech. of protein translocation in P. t. So “currently available SP prediction (SPP) programs” -- > “these 4 SP prediction programs” should be used to tone down the conclusion.

Authors’ Reply: This point of the Reviewer is well taken. The above mentioned lines have been modified as suggested by the Reviewer.

(please see page 1, lines 53-55 of the revised manuscript)

Comment 3: Authors should include the latest methods e.g. How many software are available. Why did not use latest ones e.g. DeepSig. What is rationale of using these 4 software.

Authors’ Reply: This point of the Reviewer is well taken. In the revised manuscript the information regarding the available SPP programs and the rationale of using these four SPP programs has been included.

(please see page 4, lines 101-109 of the revised manuscript)

Comment 4: # Please make suppl. table or excel showing how many SPPs are based on archaea datasets to identify archaea SPs. Provide the very briefly basic details of the methods along with the data sample size especially archaea data used for machine learning model development.

Authors’ Reply: The authors would like to present that to the best of our knowledge, only two SPPs, PRED-SIGNAL and SignalP5 are based on archaeal datsets. The basic details of these these SPPs are included in the revised manuscript.

(please see page 6, lines 170-177 of the revised manuscript)

Comment 5: # Please provide some quantitative manner to show the “Efficacy of signal peptide predictors in identifying signal peptides”, e.g. %age of proteins found to have SP using 4 software .Additionally, authors should provide for all full 3 sets in supplementary file the prediction score of all 4 software.

Authors’ Reply: This point of the Reviewer is well taken. The efficacy of SPPs has been presented in a quantitative manner. Also, the prediction scores of each SPP for full 3 sets has been presented as a supplementary file.

(please see page 9, lines 261-264 and supplementary Table 2 of the revised manuscript)

Comment 6: # Authors should order these SPPs based on some rationale. E.g. year of publish or performance? and be consistent in describing these methods all over in that order only.

Authors’ Reply: This point of the Reviewer is well taken. The SPPs have been described in a consistent order throughout the manuscript.

Comment 7: # on line 353, this thing is counter intuitive.

"The supplementary information enlisted in PRED-SIGNAL indicated that 86 proteins of P. torridus were secretory proteins (http://bioinformatics.biol.uoa.gr/PRED-SIGNAL-results/). However, when the 30 experimentally derived secreted proteins were submitted to PRED-SIGNAL it identified signal peptides in only eight proteins, while trans membrane segments were identified in five proteins." Were your 30 sequences among 1535 seq mentioned on that page of signal-pep? If yes, then how only 8 are shown to have SPs? What input was given when you say "when the 30 experimentally derived secreted proteins were submitted to PRED-SIGNAL"? Was the sequence of your 30 proteins and their 30 out of 1535 proteins different or identical? Authors need to explain this section.

Authors’ Reply: This point of the Reviewer is well taken. This section has been re-written in the revised manuscript. Hope it is clear and easy to understand. 

(please see page 12, lines 337-345 of the revised manuscript)

Comment 8: # line 80, soluble in what? Water or lipid?

Authors’ Reply: souble in water

Comment 9: # For Ref 17, what data was used to make prediction model. It is based on archaeal protein.?

Authors’ Reply: The PRED-SIGNAL was based on archaeal secretory proteins.

Comment 10: # Line 116, is there any reference for such protocol used in this section, authors may cite that.

Authors’ Reply: No reference is available for citing with protocol used for concentration of culture filtrate proteins.

Comment 11: # protein pilot software is not mentioned in the text while suppl. table 1 mentions it.

Authors’ Reply: The authors are thankful to the Reviewer for pointing this mistake. The authors have re-written these lines in the revised manuscript.

(please see page 5, lines 139-140 of the revised manuscript)

Comment 12: # Why CPU time and rates are missing for other rows in excel sheet Speed and Distribution Analysis in suppl file 1, sheet 1

Authors’ Reply: The authors would like to mention that the supplementary files containing the LC MS raw data were huge and it was difficult to upload them. During reduction of their size, the sheets containing the Speed and Distribution Analysis might have been missed. We have again uploaded the suppl files containing all the missing data.

Comment 13: # Why Global FDR is recommended at 5 and 10% unlike Global FDR fit at 1%? In protein FDR Summary sheet in supp. Tables

Authors’ Reply: The authors are thankful to the Reviewer for pointing this mistake. The global FDR fit that was used was 1% as was shown in suppl Tables. The error has been rectified in the revised manuscript.

(please see page 5, line 141 of the revised manuscript)

Comment 14: # Authors should mention version of all software and databases used in the study. Which version of backend database was used for blast.

Authors’ Reply:This point of the Reviewer is well taken. In the revised manuscript, the version of all the software and databases have been mentioned.

Comment 15: # What is meaning of % Coverage (95) and Peptides (95%) in table 1 should be mentioned.

Authors’ Reply: Since, the % Coverage (95) and Peptides (95%) is already mentioned in Supplementary file, it has been removed from Table 1. 

Comment 16: # Authors should sort table 1 based on some criteria, so that it is easy for readers to comprehend.

Authors’ Reply: This point of the Reviewer is well taken. The Table 1 contains only the details of the 30 proteins identified by LC MS. The information regarding the SPPs has been removed from these table and presented in a separate table, Table 4.

Comment 17: # Line 178: Pfam 32.0 failed to predict any domain in the hypothetical protein Q6L265? Did not find any protein with this name Q6L265. Also, what is the reason for not finding any domain?

Authors’ Reply: The authors would like to bring to your kind notice that the mention that the protein accession no. is Q6L2C5 and not Q6L265. The authors feel that the database of Pfam 32.0 probably did not contain the information about the domains in this protein may be that’s why they did not predict any domains in this protein. 

Comment 18: # Table 1 is least described in the text. Expand it like authors explain table 2. Table 1 is explained rather in discussion.

Authors’ Reply: This point of the Reviewer is well taken. As suggested by the Reviewer, the authors have described the Table 1 in detail.

(please see page 7, lines 183-190 of the revised manuscript)

Comment 19: Table 1 contains 30 proteins while fig 1 contains 34?

Authors’ Reply: The authors would like to mention that some secretory proteins were multifunctional (like malate dehydrogenase), thus they have been represented more than once. 

Comment 20: # Order fig 1 based on counts.

Authors’ Reply: This point of the Reviewer is well taken. The Fig. 1 has been reordered based on the counts.

Comment 21: # Fig 2, reorder based on counts. Also, the legend title is ‘Function’ for all the panels, it should be corrected.

Authors’ Reply: This point of the Reviewer is well taken. The Fig. 2 has been reordered based on the counts.The legend tile has also been corrected.

Comment 22: # line 226 "Here, due to the limitations of the Cytoscape software tool, we have shown only one pathway of multi-functional protein(s)". It is not clear.

Authors’ Reply: These lines have been re-written in the revised manuscript.

(please see page 9, lines 251-255 of the revised manuscript)

Comment 23: # Heading in lines 232/233 seem not in continuation. Please double check.

Authors’ Reply: The authors have corrected the headings in the revised manuscript.

Comment 24: # Fig 3, Text mentions protein id while figure contains gene ids, need to be consistent for proper interpretation of the figure. Mention about the size of nodes, length/thickness of edges if they mean anything or not.

Authors’ Reply: This point of the Reviewer is well taken. For proper interpretation of the figure the gene ids and the protein ids of the all the proteins have been included in Table 1. Also the meaning of the various symbols have been described in detail. 

(please see page 9, lines 251-255 of the revised manuscript)

Comment 25: # Line 237. Be consistent in metnioning version name e.g. InterPro 74 and Pfam 32

Authors’ Reply: This point of the Reviewer is well taken. In the revised manuscript, The authors have ensured consistency in the version names of InterPro 74 and Pfam 32.

Comment 26: # line 238 : "InterPro 74 and Pfam 32 could not find any domain, while CDD search predicted presence of a domain of Reo_sigmaC super family". But table 1 shows interpro finds a domain?

Authors’ Reply: The authors are thankful to the Reviewer for pointing this mistake. This mistake has been corrected in the revised manuscript.

(please see page 7, lines 195-197 of the revised manuscript)

Comment 27: # Section of text in 235-265 should also be available as an additional table or suppl table 

Authors’ Reply: Authors would like to present that detailed information regarding the text presented in these lines is separately presented as Tables 1 and 2.

Comment 28: # Suppl files 1,2,3 need to be explained properly as they contain multiple sub sheets. Need to explain at least one suppl file with the related content and interpretation.

Authors’ Reply: Authors would like to present that information contained in Suppl files 1,2,3 follows the common jagaron used in LC MS experiments. 

Comment 29: # Line 297/298: “…proteins might have been missed in this study from proteomic identification, due to technical constraints like, detection limit of mass spectrometry.” What is that limit in this study’s experiment.?

Authors’ Reply: The limit in this study’s experiment is the detection limit of LC MS which is in picograms.

Comment 30: # English needs to be improved including semantic and grammatical errors e.g.

in line 85 sp. Vs spp.

artifacts vs technical artefacts,

have read and approve -- > have read and approved

Authors’ Reply: This point of the Reviewer is well taken. In the revised manuscript being submitted, the authors have tried their best to remove all the semantic and grammatical errors.

---

## [Decision Letter · Decision Letter 1]

26 Jul 2021

Efficacy of signal peptide predictors in identifying signal peptides in the experimental secretome of Picrophilous torridus, a thermoacidophilic archaeon

PONE-D-21-06360R1

Dear Dr. Goel,

We’re pleased to inform you that your manuscript has been judged scientifically suitable for publication and will be formally accepted for publication once it meets all outstanding technical requirements.

Kind regards,

Dinesh Gupta

Academic Editor

PLOS ONE

Additional Editor Comments (optional):

Reviewers' comments:

Reviewer's Responses to Questions

**Comments to the Author**

1. If the authors have adequately addressed your comments raised in a previous round of review and you feel that this manuscript is now acceptable for publication, you may indicate that here to bypass the “Comments to the Author” section, enter your conflict of interest statement in the “Confidential to Editor” section, and submit your "Accept" recommendation.

Reviewer #1: All comments have been addressed

2. Is the manuscript technically sound, and do the data support the conclusions?

Reviewer #1: Yes

3. Has the statistical analysis been performed appropriately and rigorously? 

Reviewer #1: N/A

4. Have the authors made all data underlying the findings in their manuscript fully available?

Reviewer #1: Yes

5. Is the manuscript presented in an intelligible fashion and written in standard English?

Reviewer #1: Yes

6. Review Comments to the Author

Reviewer #1: (No Response)

7. PLOS authors have the option to publish the peer review history of their article (what does this mean?). If published, this will include your full peer review and any attached files.

Reviewer #1: No

---

## [Editor Report · Acceptance letter]

29 Jul 2021

PONE-D-21-06360R1 

Efficacy of signal peptide predictors in identifying signal peptides in the experimental secretome of *Picrophilous torridus*, a thermoacidophilic archaeon 

Dear Dr. Goel:

I'm pleased to inform you that your manuscript has been deemed suitable for publication in PLOS ONE. Congratulations! Your manuscript is now with our production department. 

Kind regards, 

on behalf of

Dr. Dinesh Gupta 

Academic Editor

PLOS ONE